# Beyond IbMYB1: Identification and Characterization of Two Additional Anthocyanin MYB Activators, IbMYB2 and IbMYB3, in Sweetpotato

**DOI:** 10.3390/plants14182896

**Published:** 2025-09-18

**Authors:** Jian Wang, Zhuo Chen, Yang Lu, Xiaobei Zhang, Yizhao Chen, Xiangrui Li, Yi Liu, Yonghua Liu, Sunjeet Kumar, Zhixin Zhu, Guopeng Zhu

**Affiliations:** 1School of Breeding and Multiplication (Sanya Institute of Breeding and Multiplication), Hainan University, Sanya 572025, China; 2Key Laboratory for Quality Regulation of Tropical Horticultural Crops of Hainan Province, School of Tropical Agriculture and Forestry, Hainan University, Haikou 570228, China; 3Jingmen Academy of Agriculture Science, Jingmen 448000, China

**Keywords:** sweetpotato, anthocyanins, MYB activator, IbMYB1, IbMYB2, IbMYB3, regulatory loop

## Abstract

Sweetpotato displays diverse purple pigmentation due to anthocyanin accumulation. While current research on the underlying MYB activators has focused on IbMYB1 in purple-fleshed tubers, the color diversity suggests the involvement of other MYB activators. We previously identified IbMYB2 and IbMYB3 in leaf coloration. Here, we explored the chromosomal localization, phylogeny, and evolutionary scenario of IbMYB1/2/3 using four *Ipomoea* genomes. IbMYB1/2/3 are located adjacently as an anthocyanin MYB gene cluster, likely resulting from tandem duplications. All three IbMYBs induced anthocyanins in tobacco and activated the promoters of the key anthocyanin pathway genes *IbCHS-D* and *IbDFR-B*. Expression analysis across sweetpotato varieties indicated that IbMYB1 dominates purple tuber flesh, whereas IbMYB2/3 contribute to leaf and tuber skin coloration. Overexpression of IbMYB1/2/3 in sweetpotato all induced purple fibrous roots. Transcriptomics of IbMYB2-OX fibrous roots revealed upregulation of the entire anthocyanin pathway genes. Among the most highly upregulated transcription factors were IbMYB27 and IbHLH2. An inhibitory effect induced by IbMYB27 likely accounts for the faint pigmentation in IbMYB2-OX storage roots. The role of IbMYB2/3 in fine-tuning sweetpotato’s purple pigmentation was highlighted. This study supplements previous work on IbMYB1, providing valuable insights into the intricate anthocyanin regulatory network and supporting sweetpotato breeding efforts for improved nutritional and aesthetic qualities.

## 1. Introduction

Sweetpotato (*Ipomoea batatas* L.), the world’s sixth-most cultivated crop, possesses a complex hexaploid genetic background with 90 chromosomes (2n = 6x = 90) [1,2]. It is primarily grown for its underground tubers [1,2], and some varieties are also consumed as nutrient-rich vegetables using their stems and leaves [3,4]. Anthocyanins, in addition to functioning as plant pigments, are widely recognized as among the most potent plant antioxidants [5]. Foods rich in anthocyanins offer numerous health benefits and are gaining increasing attention. For instance, purple corn has shown high economic value in the corn industry [6], and roses—traditionally ornamental—are now being used in functional foods due to their anthocyanin content [7]. Anthocyanin accumulation in sweetpotato tissues further enhances sweetpotato’s nutritional value. Purple-fleshed sweetpotato, for example, has been shown to help prevent cardiovascular diseases and various tumors [8,9,10]. In “juvenile red fading” sweetpotato leaves, which transition from red to green, anthocyanin content is strongly correlated with antioxidant capacity [3]. Understanding the molecular mechanisms behind anthocyanin accumulation in sweetpotato is crucial.

Anthocyanins are synthesized from 4-coumaroyl-CoA through a multistep enzymatic cascade involving chalcone synthase (CHS), chalcone isomerase (CHI), flavanone-3-hydroxylase (F3H), flavonoid-3′-hydroxylase (F3’H), dihydroflavonol reductase (DFR), anthocyanidin synthase (ANS), UDP-glucose flavonoid 3-glucosyltransferase (3GT), etc. [11]. As a specialized branch of flavonoid biosynthesis, anthocyanin production shares upstream enzymes (e.g., CHS) with other flavonoid branches, while downstream enzymes (e.g., DFR and 3GT) are more specific to anthocyanin formation [11,12]. In plants, transcriptional regulation of this pathway is primarily controlled by the MYB–bHLH–WD40 (MBW) protein complex. Within this ternary complex comprising three types of transcription factors (TFs), MYB has been demonstrated to play a pivotal role [13,14]. Well-known anthocyanin MYB activators include AtPAP1 in *Arabidopsis thaliana* [15], PhAN2 in petunia [16], and MdMYB10 in apple [17]. Ectopic expression of these MYBs alone can induce anthocyanin accumulation. Insights into MYB’s role as a hub can be summarized as follows: Upon activation by upstream signals, the MYB protein induces expression of its bHLH partner and then assembles with constitutively expressed WDR protein to form the MBW complex, thereby turning on the anthocyanin pathway [12,16,18].

Current research on anthocyanin MYB activators in sweetpotato has focused on IbMYB1, or more precisely, its isoform IbMYB1-2. Initially, Mano et al. [19] cloned IbMYB1 from the purple-fleshed cultivar “Ayamurasaki” and found it highly expressed in purple storage roots. They also cloned four isoforms with ~99% coding sequence (CDS) identity to IbMYB1, naming them IbMYB2s (IbMYB2-1 to IbMYB2-4). However, the so-called IbMYB2 isoforms were scarcely expressed [19]. Later, Tanaka et al. [20] cloned additional IbMYB1 isoforms (IbMYB1-1, IbMYB1-2a, and IbMYB1-2b) from an “Ayamurasaki” mutant and found that the IbMYB1-1 promoter was dysfunctional. Zhang et al. [21] further confirmed via genome-wide association studies that IbMYB1-2 is the key regulator of anthocyanin biosynthesis in sweetpotato storage roots.

However, the MYB activators responsible for anthocyanin production in other tissues—such as stems, leaves, or flowers—remain unclear. In our previous study, we cloned three distinct MYB activators from the “juvenile red fading” leaves of the cultivar “Chuanshan Zi” [18]. These were named IbMYB1, IbMYB2, and IbMYB3 after their homologs in the well-studied ornamental morning glories *I. nil* and *I. purpurea* from the genus *Ipomoea* [22]. Their CDS identities are only about 60% similar to each other. Notably, IbMYB1 is 100% identical to IbMYB1-2 and ~99% identical to the IbMYB2 isoforms reported by Mano et al. [19] and Tanaka et al. [20]. In morning glories, MYB1 alone determines flower color [12,22], while MYB2 and MYB3 remain poorly characterized and were only proposed as putative regulators in a recent genome-wide analysis of *I. nil* [23]. In sweetpotato, Zhang et al. [24] confirmed the genomic presence of IbMYB2 and various IbMYB1 isoforms across multiple cultivars with purple leaves and storage roots. However, their primary focus has been on the correlation between IbMYB1 isoforms and the purple pigmentation pattern. The role of a locus containing IbMYB1 in stem pigmentation was also suggested via OutcrossSeq [25]. Ongoing research has continued to concentrate on the purple tuberous roots and has reaffirmed the role of IbMYB1 as the key corresponding factor [26,27].

It is unlikely that IbMYB1 alone governs the diverse anthocyanin patterns across sweetpotato tissues. Further investigation into other MYB activators is warranted. In this study, we identified and characterized two additional anthocyanin MYB activators—IbMYB2 and IbMYB3—using chromosomal localization, phylogenetic and evolutionary analysis, functional verification, expression profiling, and in vivo transformation. These three IbMYBs are clustered in the genome and all function in anthocyanin induction. We analyzed their expression patterns across various cultivars and performed transcriptomic analysis of IbMYB2-OX fibrous roots, suggesting intricate regulatory mechanisms behind color patterning. Our findings offer new insights into the anthocyanin regulatory network in sweetpotato, supporting breeding efforts for improved nutritional and aesthetic traits.

## 2. Results

### 2.1. Chromosomal Distribution of IbMYB1, IbMYB2, and IbMYB3 in the Sweetpotato Genome

Three anthocyanin-related SG6 subfamily MYB activators—IbMYB1, IbMYB2, and IbMYB3—were previously identified from the R2R3-MYB family using transcriptomic data from purple sweetpotato leaves [18]. Using the cloned CDS sequences, we analyzed their similarity and physicochemical properties (Figure 1A, Appendix A). Although the three MYBs show high similarity within the R2R3 region, their C-termini differ considerably (Appendix A), with protein sequence identities ranging from 59.6% to 69.5% (Figure 1A). Their physicochemical properties also vary (Appendix A), confirming that they are distinct proteins.

The chromosomal distributions of *IbMYB1/2/3* differ across four haplotype-resolved sweetpotato genome versions. For instance, *IbMYB2* and *IbMYB3* were absent from the haplotype consensus of “Beauregard” and “Tanzania” but were located adjacent to *IbMYB1* in the “New Kawogo” and “Taizhong 6” haplotype consensus assemblies. To clarify these discrepancies, we referred to the genomes of three diploid *Ipomoea* species: *I. trifida*, *I. triloba*, and *I. nil*. In all these species, the homologs of *IbMYB1/2/3* were found adjacent to each other in clusters, with additional MYB-like loci nearby.

To better visualize these gene clusters, we examined the flanking genes of *IbMYB1/2/3* (Appendix A). We used GDP-L-galactose phosphorylase 1-like (*VTC2*) and Protein C2-DOMAIN ABA-RELATED 4 (*CAR4*) as anchor genes to define cluster boundaries. All genes within these clusters are displayed in Figure 1B and summarized in synoptic diagrams (Figure 1C).

The anthocyanin *MYB* gene clusters are located on Chr5/LG5 in *I. nil* and “Taizhong 6” but on Chr12 in *I. trifida*, *I. triloba*, and other sweetpotato haplotype genomes—reflecting the choice of reference genomes while annotating. Both *IbMYB2* and *IbMYB3* correspond to single loci with high similarity (>90%) across the three diploid *Ipomoea* genomes, closely resembling their status in sweetpotato cv. “New Kawogo” and the reannotated “Taizhong 6” (Appendix A).

Interestingly, *IbMYB1* corresponds to a single locus in *I. nil* (INIL05g09652), but no direct ortholog was identified in *I. trifida* or *I. triloba*. Instead, it aligned most closely with *IbMYB2* homologs (~81% identity; Itf12g04080.t1 and Itb12g04290.t1). In “New Kawogo”, *IbMYB2* is duplicated into two loci. In “Taizhong 6”, a segmental inversion duplication occurred within half of the gene cluster containing *IbMYB1* and two flanking *MYB2*-like genes. This resulted in two IbMYB1 loci (IB05G04610 and IB05G04780) whose CDS sequences differ by only one non-synonymous nucleotide (Appendix A; Appendix A). The genomes of *Ipomoea batatas* cv. “New Kawogo” and the reannotated “Taizhong 6” were used for further analysis.

### 2.2. Phylogeny and Evolutionary Scenario of IbMYB1, IbMYB2, and IbMYB3

Phylogenetic analysis was conducted on all MYB protein sequences within the clusters of the four *Ipomoea* species (Figure 2A, Appendix A). Sequences from the reannotated “Taizhong 6” genome were used, along with flavonoid/anthocyanin-related MYBs from *A. thaliana* and *Petunia hybrida*, to assist in classification (Appendix A).

In the phylogenetic analysis, all *Ipomoea* MYBs in the tree grouped within the anthocyanin clade and diverged into three distinct branches: MYB1, MYB2, and MYB3 (Figure 2A, Appendix A). The MYB3 branch contains homologs of IbMYB3 from all four *Ipomoea* species. The MYB2 branch is subdivided into MYB2 and MYB2-like groups. The MYB1 branch includes two members from *I. batatas* and two from *I. nil*.

The protein sequence of our cloned IbMYB1 is 100% identical to the four previously reported IbMYB1 isoforms (IbMYB1, IbMYB1-1, IbMYB1-2a, and IbMYB1-2b) [19]. In contrast, the identified IbMYB1 homologs (IB05G04610 and IB05G04780) share identical protein sequences with IbMYB2 isoforms [20] (Appendix A). Phylogenetic relationships across all three branches indicate that *I. batatas* is most closely related to *I. trifida* and most distantly related to *I. nil*.

The approximate evolutionary divergence times were calculated from Ka/Ks ratios, using the mutation rate of *CHS* in *A. thaliana*. It was estimated that IbMYB2 and IbMYB3 diverged approximately 22.73 million years ago (MYA), while IbMYB1 diverged from IbMYB2 around 17.90 MYA (Appendix A). By integrating the gene tree (Figure 2A), Ka/Ks ratios (Appendix A), and physical gene positions within clusters (Figure 1B,C), we propose a hypothetical evolutionary scenario (Figure 2B, Appendix A): An ancestral anthocyanin MYB likely underwent tandem duplications, giving rise to three MYB genes (MYB3, MYB2, and one MYB2-like), as observed in *I. trifida* and *I. triloba*. IbMYB1 appears to have evolved from a MYB2-like gene around 10.29 MYA. Further tandem duplications then led to the formation of the cluster in “New Kawogo”, while in “Taizhong 6”, an additional segmental inversion duplication subsequently occurred.

### 2.3. Functional Characterization of IbMYB1, IbMYB2, and IbMYB3 in Anthocyanin Induction

We functionally characterized IbMYB1, IbMYB2, and IbMYB3 using subcellular localization, transient expression assays in tobacco leaves, and dual-luciferase promoter activation assays (Figure 3).

All three MYB proteins were found to co-localize with the nuclear marker mKATE (Figure 3A). In tobacco leaves, red pigmentation appeared around the infiltration sites within four days after overexpression of each IbMYB (Figure 3B). Among IbMYB1/2/3, IbMYB3 exhibited the strongest anthocyanin induction, as demonstrated by repetitions of the assay.

In the promoter activation tests, the promoters of the upstream gene *IbCHS-D* and the downstream gene *IbDFR-B*, which are representatives of the anthocyanin structural genes, were utilized. When MYB was absent, the joint TF injection of IbbHLH2 and IbWDR1 could not activate the promoters. Sole TF injection of IbMYB1, IbMYB2, or IbMYB3 slightly induced *pIbCHS-D* and moderately activated *pIbDFR-B*. The co-injection of either IbMYB1, IbMYB2, or IbMYB3 with IbbHLH2 and IbWDR1, which would form the MBW complex, significantly enhanced promoter activation (Figure 3C).

### 2.4. Expression of IbMYB1, IbMYB2, and IbMYB3 in Representative Sweetpotato Varieties

Nine sweetpotato varieties with distinct leaf and storage root colors were analyzed (Figure 4A). Among them, “Fu18” and “HD-V7” have green leaves, while “Golden” has yellow leaves with red edges. The varieties “Ayamurasaki”, “Jinghong6”, and “HD7791” exhibit reddish young leaves that turn green upon maturation, with purple, orange, and ivory-white tuber flesh, respectively. “Xu1402”, “Bei5”, and “Chuzi1” display greenish young leaves with red veins and reddish mature leaves (Figure 4A, Appendix A). Tuber flesh colors vary from ivory to yellow, orange, and purple among the nine varieties. The purple-fleshed tubers accumulate anthocyanins, as seen in “HD-V7” and “Ayamurasaki”; whereas the orange-fleshed tubers, as seen in “Jinghong6”, accumulate carotenoids.

Expression analysis focused on four regulatory genes (*IbMYB1/2/3* and *IbbHLH2*) and three structural genes (*IbCHS-D*, *IbDFR-B*, and *Ib3GT*) spanning from upstream to downstream of the anthocyanin pathway (Figure 4B, Appendix A). Due to distinct nucleotide sequences, all three IbMYBs showed high amplification specificity in qRT-PCR (Appendix A).

The variety “Fu18”, which lacks anthocyanin accumulation throughout the entire plant, showed minimal expression of *IbMYB1/2/3*. However, moderate levels of *IbCHS-D* and *IbDFR-B* were detected, possibly due to the expression of other colorless flavonoids. In contrast, other varieties display varied patterns of anthocyanin accumulation.

The expression patterns of *IbMYB1/2/3* among varieties can be broadly summarized as follows:(1)Purple tuber flesh correlates strongly with the expression of *IbMYB1*, for instance, in the varieties “HD-V7” and “Ayamurasaki”.(2)Purple tuber skin is influenced mainly by *IbMYB1* alone or in combination with *IbMYB3*.(3)Purple leaves result from the expression combination of *IbMYB1*, *IbMYB2*, and *IbMYB3*. The combination included *IbMYB1* (e.g., HD7791), *IbMYB1/2* (e.g., Ayamurasaki), *IbMYB1/3* (e.g., Chuzi1), *IbMYB2/3* (e.g., Jinghong6), and *IbMYB1/2/3* (e.g., Bei5).

Generally, *IbMYB1* was expressed in nearly all purple tissues and is the most significant determinant of anthocyanin accumulation across all four tissues. “Bei5” exhibits high expression of all three *IbMYBs* in leaves. “Jinghong6” shows minimal expression of *IbMYB1*, with *IbMYB2/3* driving anthocyanin production, and its green mature leaves display significantly reduced yet still noticeable gene expressions.

The expression of the other four genes (*IbbHLH2*, *IbCHS-D*, *IbDFR-B*, and *Ib3GT*) generally correlates with anthocyanin accumulation; however, some discrepancies may arise due to their involvement in other flavonoid pathways that share structural genes with the anthocyanin pathway.

### 2.5. Overexpression of IbMYB1, IbMYB2, and IbMYB3 inSweetpotato Roots

The widely cultivated vegetable-type variety “Fu18”, which lacks anthocyanin accumulation, was used for in vivo overexpression of IbMYB1/2/3 via *A. rhizogenes*-mediated root transformation. Compared to the white fibrous roots produced by the control group (CK) transformed with the empty vector, overexpression of IbMYB1/2/3 (IbMYB1-OX, IbMYB2-OX, and IbMYB3-OX) all induced purple fibrous roots (Figure 5A). Among 60 transformed vines per group, purple roots appeared in 50 IbMYB1-OX and 54 IbMYB2-OX vines, while only 5 IbMYB3-OX vines exhibited pale purple roots.

The IbMYB2-OX plants grown in vermiculite developed enlarged tuberous roots within 40 days. However, the positive storage roots only exhibited reddish color on the skin; the interior flesh remained white, in contrast to the light yellow flesh of the CK group (Figure 5B). This result suggests that anthocyanin accumulation may disrupt the biosynthesis of other yellow flavonoids.

### 2.6. Transcriptomic Analysis of IbMYB2-Overexpressing Fibrous Roots

Transcriptomic analysis was conducted on fibrous roots of two groups: the empty vector transgenic control group (CK) and the IbMYB2 overexpression group (IbMYB2-OX). A total of 8061 differentially expressed genes (DEGs) were identified, with 4930 upregulated and 3131 downregulated (Figure 6A). Among the top 20 enriched KEGG pathways, four were directly associated with anthocyanin biosynthesis: phenylpropanoid biosynthesis (ko00940, including PAL, C4H, and 4CL), flavonoid biosynthesis (ko00941, including CHS, CHI, F3H, F3’H, DFR, and ANS), anthocyanin biosynthesis (ko00942, including 3GT and 3GGT), and flavone and flavonol biosynthesis (ko00944, including F3’H) (Figure 6B).

Key anthocyanin-related regulatory and structural genes were identified (Appendix A), and the most significant members were selected based on transcripts per million (TPM) and log2(FC). These genes were visualized as upregulated in the volcano plot (Figure 6C) and pathway heatmap (Figure 6D) and further validated by qRT-PCR (Figure 6E). Along with the overexpressed IbMYB2, all anthocyanin structural genes and IbbHLH2 were upregulated. In contrast, IbWDR1 expression remained unchanged, consistent with the constitutive expression pattern of WDR members in the MBW complex.

We also assessed the expression of IbMYB1 and IbMYB3 (Appendix A). IbMYB1 exhibited extremely low TPM values in transcriptomic data and was undetectable in qRT-PCR assays in both the CK and IbMYB2-OX groups. Similarly, IbMYB3 showed low TPM values, with no significant difference between the two groups.

### 2.7. Screening of Highly Associated TFs in IbMYB2-OX Fibrous Roots

Although the anthocyanin pathway is primarily regulated at the transcriptional level by the MBW complex, it is also modulated by numerous other TFs. From the DEG list of TFs, we selected the top 50 up- and downregulated genes. After filtering out genes with low TPM values or redundant annotations, we identified 13 upregulated and 14 downregulated TFs (Figure 7A, Appendix A).

Among the 13 upregulated TFs, five belong to the MYB family, four to the bHLH family, three to the ERF family (DREB1E, ERF109, and ERF025), and one to the WRKY family (WRKY27) (Figure 7A, Appendix A). The most highly upregulated TFs were IbMYB2, IbMYB27, and IbbHLH2. The four bHLH genes, originally annotated as BHLH42 based on *Arabidopsis* AtTT8, were confirmed to correspond to IbbHLH2, the known anthocyanin bHLH regulator in *Ipomoea* (Appendix A). The remaining two MYBs had similar symbols or descriptions to IbMYB27 and were classified as SG16 and SG17 MYB members. The 14 downregulated TFs represented multiple families, with three from HSF, three from NAC, three from TALE, and one each from five other families.

Family enrichment analysis revealed four significantly enriched TF families: WRKY, ERF, GRAS, and HSF (Figure 7B). These families are primarily linked to stress or hormone responses. Most DEGs in WRKY, ERF, and GRAS were upregulated, while the majority in HSF were downregulated. The MYB family ranked seventh, with a non-significant enrichment factor, yet included a substantial number of 37 DEGs.

In addition to *IbMYB2* and *IbbHLH2* (Figure 6E), eight additional TF genes were validated by qRT-PCR (Figure 7C), and the results agreed with the transcriptomic data.

## 3. Discussion

### 3.1. Identification of IbMYB2 and IbMYB3 in Addition to the Widely Recognized IbMYB1

The widely recognized IbMYB1 was initially identified through comparative studies of purple-fleshed and non-purple-fleshed sweetpotato [19] and has been continually reaffirmed [26,27]. Although later studies confirmed that IbMYB1 is essential for purple pigmentation in leaves and explored the relationship between its isoforms and coloration [20,24], these isoforms (IbMYB1-1, IbMYB1-2a, and IbMYB1-2b) exhibit nearly identical CDS and share >95% identity with the reported IbMYB2s. However, the genes encoding these isoforms display significant structural differences in their promoters and introns, likely due to transposon activity, which may influence their expression patterns. While IbMYB1-1 and IbMYB2s are reported as scarcely expressed [19,24], IbMYB1-2 is the main expressed isoform [27].

In contrast to the presence of multiple isoforms for IbMYB1, homologs of IbMYB2 and IbMYB3 are relatively conserved across species. The identification of IbMYB2 and IbMYB3 was based on two specific conditions: (1) the use of “juvenile red fading” leaves, which transition from purple apical leaves to mature green leaves. This classic plant phenomenon theoretically involves a complex regulatory mechanism requiring multiple TFs. (2) The application of de novo transcriptome assembly. At the time, reference genomes were incomplete [1]. De novo assembly revealed three expressed SG6 anthocyanin MYBs. Due to high CDS similarity, various IbMYB1 isoforms were merged into a single unigene, whereas IbMYB2 and IbMYB3 were successfully cloned and confirmed [18].

### 3.2. Chromosomal Localization of IbMYB1/2/3 as Cluster in Sweetpotato Genome

Multiple anthocyanin MYB genes were found within a single gene cluster in all four Ipomoea species examined. The chromosomal localization of the cluster varies depending on the reference genome. These clusters are located on Chr5/LG5 in *I. nil* and the sweetpotato cv. “Taizhong 6” but on Chr12 in *I. trifida*, *I. triloba*, and three other sweetpotato haplotype genomes. This discrepancy arises from the choice of different reference genomes during annotating: The “Taizhong 6” [1,28] annotation utilized *I. nil* genome [29], while the SweetGAINS Project assemblies used those of *I. trifida* and *I. triloba* [2]. Also, discrepancy occurred in GWAS analysis of the sweetpotato anthocyanin locus: the locus was mapped to Chr5 when referencing the “Taizhong 6” genome [25] but to Chr12 when referencing the *I. trifida* and *I. triloba* genomes [30].

Similar cluster distributions have been reported in various other plants. Xue et al. [31] identified clusters of two to six anthocyanin MYBs in eight eudicot species, including *A. thaliana*, orange, pear, grape, apple, carrot, Chinese bayberry, and *Prunus*. Genomic synteny and collinearity analyses suggest that these tandem MYBs reside within a highly conserved region across many eudicot genomes.

Functional differentiation among MYBs within a cluster mainly arises through two mechanisms: (1) protein inactivation or functional alteration due to CDS mutations and (2) altered gene expression often associated with transposon insertions in promoters or introns. For instance, in grape, a four-MYB cluster (VvMYBA1 to VvMYBA4) regulates skin color. Simultaneous mutations in VvMYBA1 (CDS mutation) and VvMYBA2 (retrotransposon in the promoter) result in a transition from red to white grape skin [32]. Similarly, Chinese bayberry contains a four-MYB cluster (MrMYB1.1, MrMYB1.2, MrMYB1.3, and MrMYB2), where specific alleles of MrMYB1.1 and exclusive expression of MrMYB1.3 contribute to fruit color variation [31]. In sweetpotato, transcriptomic data suggest that the MYB2-like gene within the cluster is not expressed. Although IbMYB1/2/3 are expressed, their functional differentiation remains unclear.

### 3.3. Expression of IbMYB1, IbMYB2, and IbMYB3 in Sweetpotato Tissues

Analysis of IbMYB1/2/3 expression in leaves and storage roots across multiple sweetpotato varieties indicates that purple flesh coloration is likely controlled exclusively by IbMYB1. Expression of IbMYB2/3 was minimal in tuber flesh of any color, consistent with previous reports [19,26,27]. In leaves and tuber skins, IbMYB2 and IbMYB3 were also involved alongside IbMYB1. In purple/red tuber skins, co-expression of IbMYB1 and IbMYB3 was common, though expression patterns involving IbMYB1 alone, IbMYB3 alone, or IbMYB2/3 were also observed. In red leaves, expression profiles of IbMYB1/2/3 varied among varieties, with combinations including IbMYB1 alone, IbMYB1/2, IbMYB1/2/3, and IbMYB2/3 all detected. Nonetheless, IbMYB1 remains the most prominent in leaves and tuber skins, as it is expressed in nearly all purple tissues. These findings align with current reports that IbMYB1 is crucial for purple pigmentation in sweetpotato tuber flesh and leaves [24].

### 3.4. Feedback Regulation by the MBW Complex Makes Expression of IbMYB1/2/3 Essential for Sustaining Purple Coloration

When overexpressed under the 35S promoter, all three IbMYBs effectively induced ectopic anthocyanin accumulation in tobacco and produced purple fibrous roots in sweetpotato. However, the purple coloration faded in subsequent tuberous roots. Analysis of differentially expressed TFs in IbMYB2-OX fibrous roots revealed that IbMYB27 and IbHLH2 were the top upregulated TFs. IbHLH2 is the bHLH member of the sweetpotato MBW complex, while IbMYB27 is an MYB repressor that competes with MYB activators to form a repressive MBW complex [18,26].

The feedback regulatory loop triggered by IbMYB27 likely explains the diminished purple pigmentation in the storage roots of IbMYB2-OX plants. The fact that only 5 out of 60 IbMYB3-OX vines displayed pale purple roots may also be attributed to this feedback loop, especially since IbMYB3 showed the strongest anthocyanin induction capability among IbMYB1/2/3 and may have potently induced IbMYB27. These results underscore the delicate balance of anthocyanin regulatory loops in sweetpotato tissues (Figure 8) and support the conclusion by Zhang et al. [24] that sustained IbMYB1 expression is necessary to maintain purple coloration. Beyond the MBW complex, hierarchical regulation by HY5 and WRKY factors has also been demonstrated [7]. Our data further suggest that TFs from the WRKY, ERF, GRAS, and HSF families may participate in this intricate anthocyanin regulatory network.

## 4. Materials and Methods

### 4.1. Plant Materials

Sweetpotato varieties were obtained from the National Tropical Plants Germplasm Resource Center and cultivated in the fields at Hainan University (Sanya, China). Nine representative varieties were selected for this study, including three for vegetable use (Fu18, HD-V7, and HD7791), three for ornamental use (Golden, Xu1402, and Chuzi1), and three for tuber use (Ayamurasaki, Jinghong6, and Bei5). Organic fertilizer was applied to the soil, and healthy vines were planted every four months from March to October between 2022 and 2024. Representative vines and tuberous roots were photographed and used for gene expression analysis. The apical leaf was collected as the young leaf (YL), while the fourth leaf from the top was collected as the mature leaf (ML). For tuberous roots, the root skin (RS) was prepared by scraping the surface with a blade, and the root flesh (RF) was obtained by chopping the inner flesh tissue. Each sample was derived from three vines or tubers pooled together, and three samples were obtained as biological replicates. The collected samples were frozen in liquid nitrogen and stored at −80 °C for subsequent qRT-PCR assays.

Tobacco plants, including *Nicotiana tabacum* for transient anthocyanin induction assays and *N. benthamiana* for subcellular localization and dual-luciferase assays, were grown in pots in growth chambers under 16 h light/8 h dark cycles at 25 °C.

### 4.2. Physicochemical Properties and Chromosomal Localization Analysis of IbMYB1/2/3

The complete CDSs of IbMYB1/2/3 were cloned from sweetpotato variety “Chuanshan Zi,” as reported by Deng et al. [18]. The physicochemical properties, including molecular weight, isoelectric point, instability index, and grand average of hydropathy, were predicted by the ProtParam tool (https://web.expasy.org/protparam/, accessed on 19 September 2024).

Chromosome localization analysis was performed for IbMYB1/2/3 using multiple versions of haplotype-resolved sweetpotato genomes, including three chromosome-scale genome assemblies from the SweetGAINS Project (*Ipomoea batatas* cv. Beauregard, Tanzania, and New Kawogo; https://sweetpotato.uga.edu/, accessed on 4 May 2025) and the reannotated genome of *Ipomoea batatas* cv. “Taizhong 6” by Liang et al. [28] (https://www.sweetpotao.com/download_genome.html, accessed on 27 September 2024). The initial “Taizhong 6” genome assembly by Yang et al. [1] was known to suffer from redundancy, incompleteness, and numerous misassemblies and was not used here. Genomes of three closely related diploid *Ipomoea* species were used to clarify the possible evolutionary history, including *I. trifida*, *I. triloba*, and *I. nil* [2,29]. Genomes of *I. trifida* and *I. triloba* were obtained from NCBI (https://www.ncbi.nlm.nih.gov/, accessed on 27 September 2024), and the *I. nil* genome was sourced from “*Ipomoea nil* Genome Project” (http://viewer.shigen.info/asagao/download.php, accessed on 27 September 2024). Homologous genes with over 90% identity were further screened based on alignment length. Positional information was extracted from the sweetpotato genome annotation and visualized using TBtools-IIv2.310 (https://github.com/CJ-Chen/TBtools-II, accessed on 27 September 2024).

### 4.3. Phylogenetic and Evolutionary Analysis of the MYBs in the Clusters

We constructed phylogenetic trees from MYB protein sequences using MEGA 11 [33]. The neighbor-joining tree was generated using the *p*-distance model with pairwise deletion and 1000 bootstraps. We analyzed evolutionary pressure and divergence times for IbMYB1/2/3 using Ka/Ks calculations. Ka and Ks represent the number of non-synonymous and synonymous mutations, respectively. A Ka/Ks ratio >1 indicates positive selection, =1 signifies neutral selection, and <1 suggests purifying selection. The approximate divergence time (T) was calculated as T = Ks/2r, where T is measured in million years ago (MYA), and r is the mutation rate. The mutation rate of the *CHS* gene from *A. thaliana* (1.5 × 10^−8^ synonymous substitutions per site per year) was utilized as a reference [34] due to the absence of clear and unified reports on the mutation rates of MYB genes or the *Ipomoea* genus.

### 4.4. Subcellular Localization and Transient Anthocyanin Induction Assays in Tobacco

The CDSs of IbMYB1/2/3 were fused to the N-terminus of GFP in the pCambia1300-GFP vector, and the resulting plasmids were introduced into *Agrobacterium tumefaciens* strain EHA105. Using *Agrobacterium*-mediated transformation, these plasmids were infiltrated into leaves of *N. benthamiana* for subcellular localization analysis and into leaves of *N. tabacum* for transient anthocyanin induction assays. The pCambia1300-GFP empty vector was used as control.

After transformation, *N. benthamiana* plants were grown under dim light for two days, and the leaf epidermis was observed for subcellular localization under a fluorescence microscope (Nikon C2-ER, Nikon, Japan). The imaging included a bright-field channel and three fluorescence channels: GFP (488/510 nm), mKATE (561/580 nm), and chlorophyll autofluorescence (640/675 nm). Here, mKATE was fused to a nuclear localization signal (NLS) and served as a nuclear marker.

For anthocyanin induction, *N. tabacum* plants were incubated in darkness for 24 h post-transformation and then returned to normal growth conditions. Leaves turned red by the third day and were photographed on the fourth day after *Agrobacterium* injection. The red color persisted until the leaves withered.

### 4.5. Promoter Activation Tests by Dual-Luciferase Assay

Promoter sequences of *IbCHS-D* (555 bp) and *IbDFR-B* (308 bp) [18] were cloned into the pGreen II-0800-Luc vector. The CDSs of IbMYB1/2/3, IbbHLH2, and IbWDR1 were inserted into the pGreen II-62-SK vector. These constructs were each transformed into *A. tumefaciens* (GV3101). Combinations of the *A. tumefaciens* strains harboring promoter and TF constructs were co-infiltrated into *N. benthamiana* leaves. After two days of cultivation under dim light, the leaves were harvested and imaged using a CCD camera (Andor Technology, Belfast, UK).

### 4.6. Agrobacterium Rhizogenes-Mediated In Vivo Root Transformation of Sweetpotato

The vegetable-type variety “Fu18” was utilized for transformation with the 35S::IbMYB1/2/3 constructs. *A. rhizogenes*-mediated in vivo root transformation system was modified from Cao et al. [35] and Zhang et al. [36]. The pCambia1300-35S::GFP empty vector and pCambia1300-35S::IbMYB1/2/3-GFP constructs were introduced into *A. rhizogenes* strain K599. Healthy vine cuttings of “Fu18” were prepared, with 60 vines per transformation group. Leaves, petioles, and adventitious roots from the basal 3–4 stem nodes were removed, and the stem nodes were wounded with a syringe. The wounded areas were immersed in the *Agrobacterium* suspension solution and incubated in the dark for 8–12 h, followed by incubation in ddH_2_O for an additional day. Antibacterial treatment was then performed by immersing the vines in 200 mg/L Carbenicillin solution for 1 h, after which the vines were rinsed twice with ddH_2_O and planted in moist vermiculite. Purple fibrous roots emerged after three weeks and were partially confirmed by PCR. The transformed purple fibrous roots, along with the white fibrous roots transformed by empty vector, were harvested and immediately frozen in liquid nitrogen for subsequent transcriptomic analysis and qRT-PCR.

### 4.7. Transcriptomic Analysis

Transcriptomic analysis was performed on two groups, with three biological replicates per group: the IbMYB2-overexpression roots (IbMYB2-OX) and the empty vector-transformed control roots (CK). The samples were transported via a dry-ice chain to Gene Denovo Biotechnology Co. (Guangzhou, China), which conducted the extraction of total RNA, the construction of cDNA libraries, and subsequent workflows. The final sequencing was performed on the Illumina Novaseq6000.

High-quality clean reads were mapped to the “Xu Shu 18” reference genome (https://plantgarden.jp/ja/list/t4120/genome/t4120.G001 (accessed on 12 January 2024)). Transcript expression levels were calculated as transcripts per million (TPM). Differentially expressed genes (DEGs) were identified with the criteria |log_2_ fold change (FC)| > 1 and false discovery rate (FDR) < 0.05. FDR correction was carried out using Benjamini–Hochberg procedure at *p*-adjust < 0.05. Bioinformatic analysis was performed using Omicsmart, a dynamic real-time interactive online platform for data analysis (http://www.omicsmart.com, accessed on 1 March 2024).

### 4.8. qRT-PCR

qRT-PCR was performed with three biological replicates. Total RNA was extracted using the RNAprep Pure Plant Plus Kit (DP432, TIANGEN, Beijing), and first-strand cDNA was synthesized with the FastKing cDNA Dispelling RT SuperMix (KR126, TIANGEN, Beijing). qRT-PCR was conducted using PerfectStart Green qPCR SuperMix (AQ601, TransGen, Beijing). The expression levels were normalized to the average Ct value of two reference genes, *Actin* and *ARF*. The cDNA template amount was adjusted so that the average Ct of these reference genes was ~22.

For cross-variety gene expression analysis, relative expression was calculated using the ΔCt method relative to the reference genes (Appendix A) [37]. Differences among tissues and varieties for each gene were evaluated by one-way ANOVA (*p* < 0.05) using IBM SPSS Statistics 27 (SPSS Inc., Chicago, IL, USA). Mean ΔCt values are visualized in a heatmap (Figure 4B). Lower ΔCt values correspond to higher expression. Genes with ΔCt < 9 were considered as expressed and those with ΔCt > 13 as not expressed. The heatmap uses a ΔCt interval of 3 (8-fold change), with color gradients roughly denoting significant expression differences.

For transcriptomic data validation, relative expression levels were calculated using the 2^−ΔΔCt^ method. Data are presented as mean ± standard error (SE) (n = 3). Significance was assessed via *t*-tests, with *p* < 0.05 (*) and *p* < 0.01 (**). All primers used are listed in Appendix A.

## 5. Conclusions

While current research on sweetpotato anthocyanin MYB activators has primarily focused on IbMYB1, this study identified and characterized two additional activators—IbMYB2 and IbMYB3—using an integrated approach that included chromosomal localization, phylogenetic and evolutionary analyses, functional verification, expression profiling, and in vivo transformation. Our results demonstrate that IbMYB1, IbMYB2, and IbMYB3 are three distinct anthocyanin activators clustered in the sweetpotato genome. All three IbMYBs effectively activated key anthocyanin biosynthesis genes and promoted ectopic anthocyanin accumulation when overexpressed in tobacco. However, their overexpression in sweetpotato may induce feedback regulation by upregulating the repressor IbMYB27. In sweetpotato tissues, purple tuber flesh coloration appears to be specifically controlled by IbMYB1, whereas all three IbMYBs contribute to pigmentation in leaves and tuber skins. This study supplements earlier work on IbMYB1 and enhances our understanding of the intricate regulatory network governing anthocyanin accumulation in sweetpotato. Further research into the molecular mechanisms underlying the functional differentiation among these three IbMYBs will provide deeper insights for breeding sweetpotato varieties with enhanced nutritional and aesthetic qualities.

## Figures and Tables

**Figure 1 plants-14-02896-f001:**
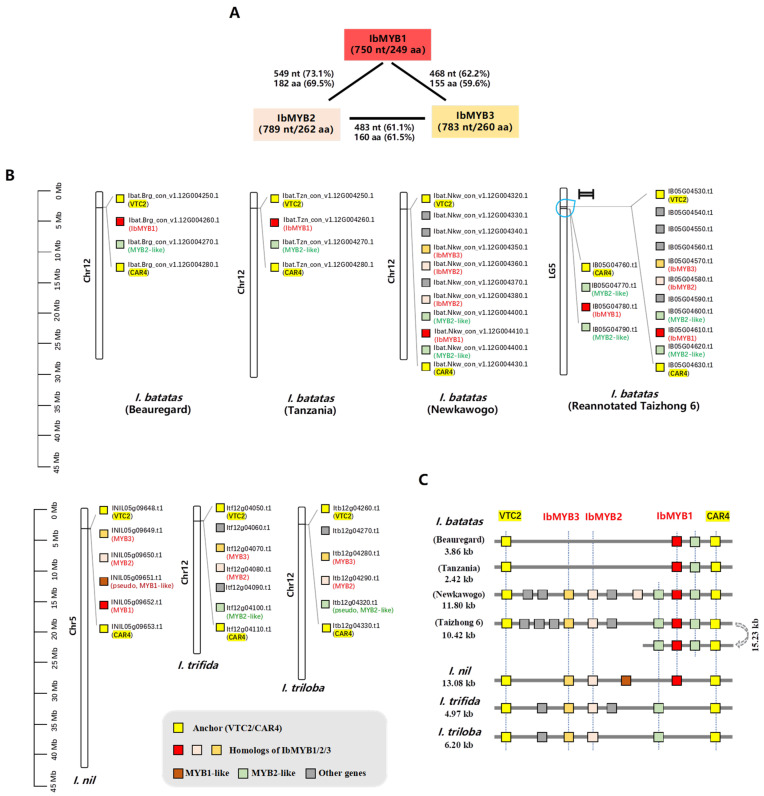
Sequence similarity and chromosomal localization of *IbMYB1*, *IbMYB2*, and *IbMYB3*. (**A**) Nucleic acid and protein sequence similarity among the CDSs of three IbMYBs. (**B**) Chromosomal distribution of anthocyanin MYB gene clusters in sweetpotato and related *Ipomoea* species. Yellow-highlighted VTC2 and CAR4 mark the conserved boundaries of each cluster. (**C**) Schematic diagrams of the anthocyanin-related *MYB* gene clusters in *Ipomoea* species. Cluster lengths are indicated. Colored blocks denote gene types: *IbMYB1/2/3* homologs (red to pink), *IbMYB1*-like (brown), *IbMYB2*-like (green), and unrelated genes (gray). Homologous genes are connected by dashed lines. A segmental inversion duplication is shown in the *IbMYB1*-containing region of *I. batatas* cv. “Taizhong 6”.

**Figure 2 plants-14-02896-f002:**
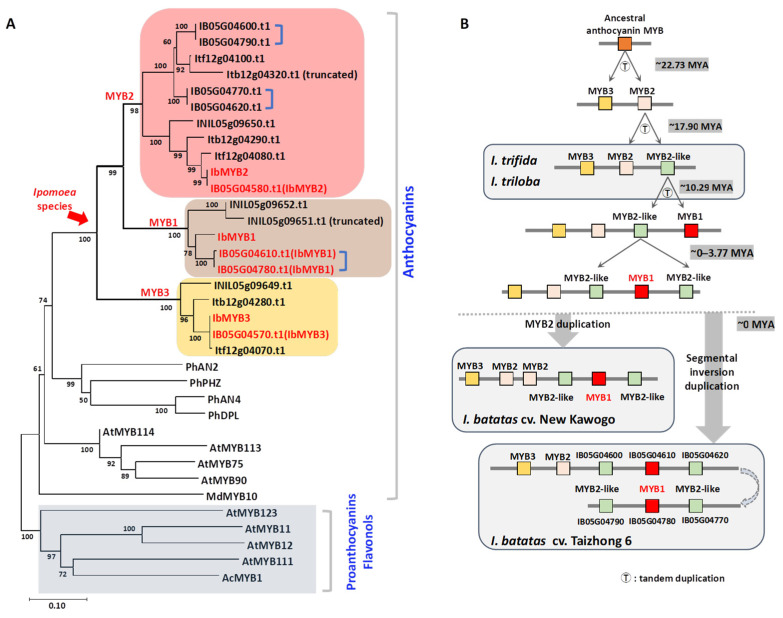
Phylogenetic relationship and evolutionary scenario of *IbMYB*s. (**A**) Phylogenetic tree of IbMYB1/2/3 and their homologs in *Ipomoea*. Flavonoid/anthocyanin-related MYBs from *Arabidopsis* and *Petunia* were used as references. The three MYB pairs resulting from segmental inversion duplication are marked by half square brackets. (**B**) Hypothetical evolutionary scenario of the IbMYB1/2/3 cluster, inferred from gene tree topology, Ka/Ks ratios, and genomic locations. “T” marks putative tandem duplication events. Shaded boxes represent the current genomic structures in *Ipomoea* species. Estimated divergence times are in million years ago (MYA).

**Figure 3 plants-14-02896-f003:**
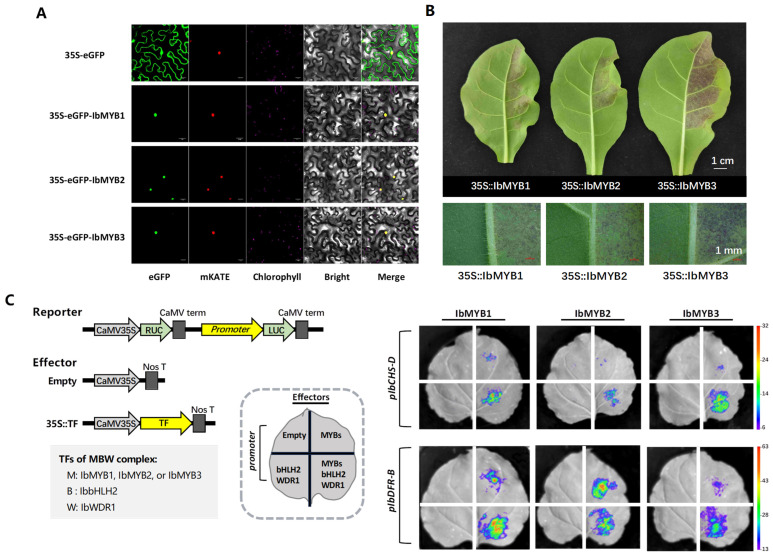
Functional verification of *IbMYB1*, *IbMYB2*, and *IbMYB3*. (**A**) Subcellular localization. mKATE was used as a nucleus marker. Bar = 10 μm. (**B**) Transient anthocyanin induction assays in *N. tabacum* leaves. Whole leaves (upper panels; bar = 1 cm) and enlarged views around the central vein (lower panels; bar = 1 mm) are shown. (**C**) Representative bioluminescence images showing the activation of the *IbCHS-D* and *IbDFR-B* promoters by IbMYB1/2/3 in tobacco leaves. Diagrams of the reporter and effector vectors and the layout of injection combinations are shown on the left.

**Figure 4 plants-14-02896-f004:**
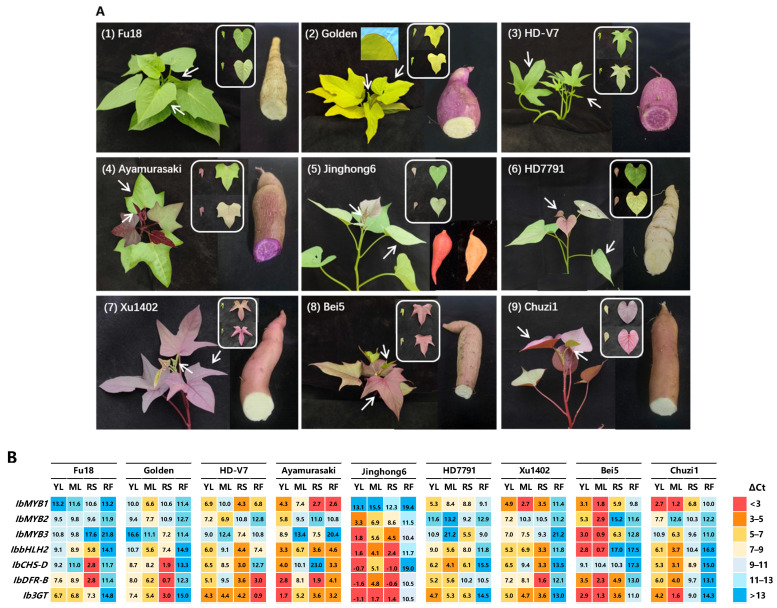
Expression analysis of *IbMYB1*, *IbMYB2*, *IbMYB3*, and key anthocyanin genes in representative sweetpotato varieties. (**A**) Phenotypes of the nine varieties. White arrows indicate the young and mature leaves sampled. For leaves, upper and lower panels show adaxial and abaxial sides, respectively. The red leaf edge in “Golden” is highlighted (inset). (**B**) Heatmap of gene expression across tissues, represented as ΔCt values relative to housekeeping genes. Lower ΔCt values signify higher expression, with a difference of 1 corresponding to a 2-fold change. Colors represent expression levels in ΔCt increments of 3 (~8-fold change). Tissues assayed include young leaf (YL), mature leaf (ML), root skin (RS), and root flesh (RF).

**Figure 5 plants-14-02896-f005:**
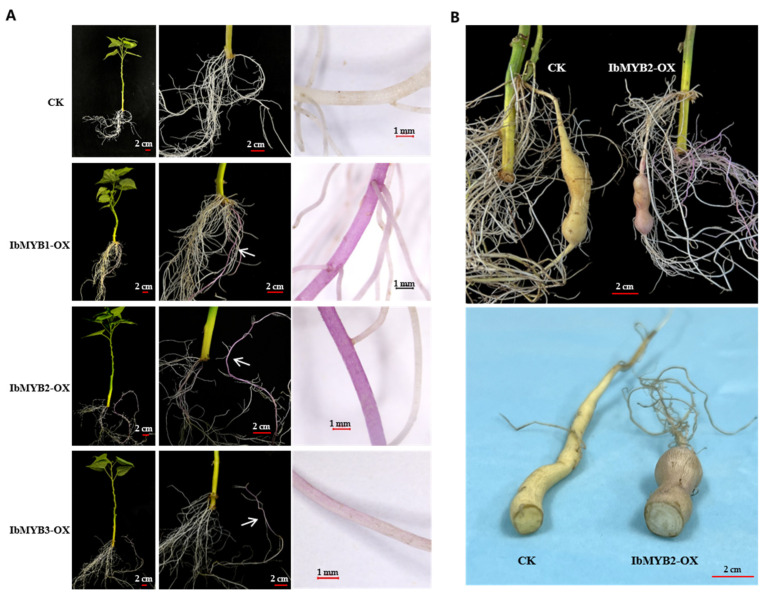
Overexpression of *IbMYB1*, *IbMYB2*, and *IbMYB3* induces anthocyanin accumulation in sweetpotato roots. (**A**) Purple fibrous roots induced by IbMYB1/2/3 overexpression. Left panels: transformed vine cuttings; bar = 2 cm. Middle panels: enlarged views of roots, with arrows indicating purple fibrous roots; bar = 2 cm. Right panels: microscopic images; bar = 1 mm. (**B**) Tuberous roots of IbMYB2-OX plants show reddish skin but white flesh, in contrast to the light yellow flesh of CK; bar = 2 cm.

**Figure 6 plants-14-02896-f006:**
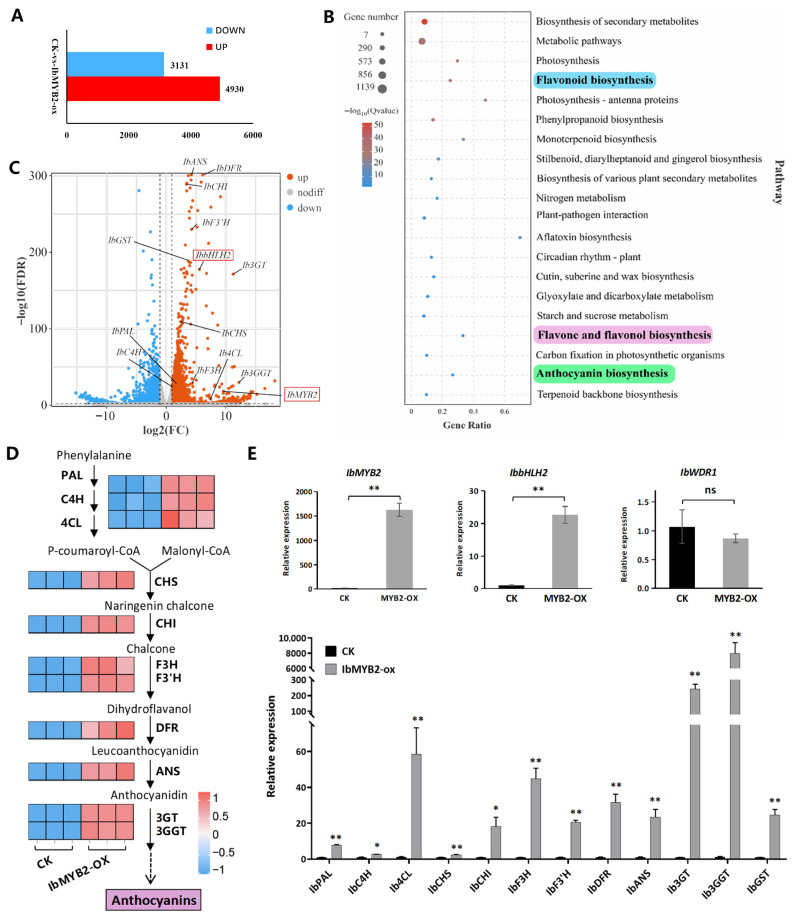
Transcriptomic analysis of DEGs in CK vs. IbMYB2-OX fibrous roots. (**A**) Number of DEGs, filtered with |log_2_FC| > 1 and FDR < 0.05. (**B**) Top 20 significantly enriched KEGG pathways. Anthocyanin-related pathways are highlighted. (**C**) Volcano plot of anthocyanin-related DEGs. Red boxes highlight IbMYB2 and IbbHLH2. (**D**) Heatmap of gene expression along the anthocyanin pathway. (**E**) qRT-PCR validation of anthocyanin structural genes and regulatory TFs (MYB, bHLH, WDR). Data are presented as means ± SE (n = 3). Significance was tested using *t*-tests (* *p* < 0.05; ** *p* < 0.01).

**Figure 7 plants-14-02896-f007:**
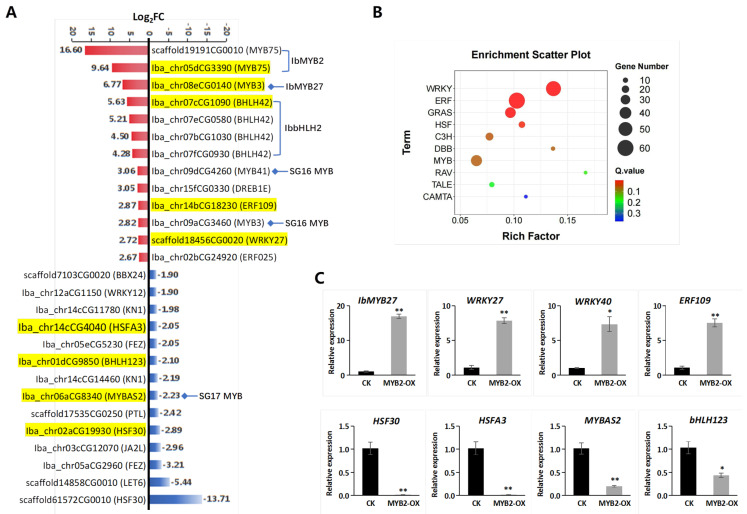
Analysis of TFs among DEGs from CK vs. IbMYB2-OX fibrous roots. (**A**) Top upregulated (red columns) and downregulated (blue columns) TFs. Gene IDs are followed by gene symbols in brackets. DEGs belonging to the MBW complex or the MYB family have been reannotated. Genes selected for qRT-PCR validation are highlighted in yellow. (**B**) Top 10 enriched TF families among DEGs. Only the top four families show significant enrichment (Q value < 0.05). (**C**) qRT-PCR validation of selected TFs. Data are presented as means ± SE (n = 3). Significance was tested using *t*-tests (* *p* < 0.05; ** *p* < 0.01).

**Figure 8 plants-14-02896-f008:**
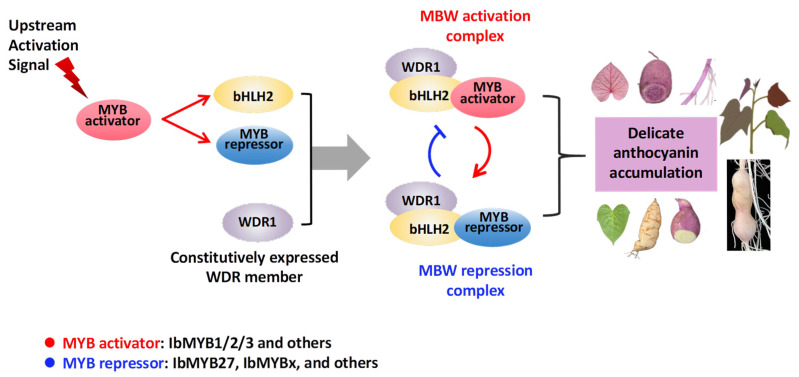
Proposed model for the intricate regulation of anthocyanin accumulation in sweetpotato tissues. This model integrates hierarchical control and feedback mechanisms. Red and blue arrows represent activation and repression, respectively. Sustained expression of MYB activators is crucial for maintaining purple pigmentation. Conversely, negative feedback mediated by MYB repressors (e.g., IbMYB27 and IbMYBx) can result in color fading, as illustrated in the right panels depicting pigment loss in developing leaves and storage roots.

## Data Availability

The RNA-Seq data are available from the Genome Sequence Archive (GSA) in the National Genomics Data Center under accession number CRA016456 (https://ngdc.cncb.ac.cn/gsa).

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
