# Peer review of "Beyond IbMYB1: Identification and Characterization of Two Additional Anthocyanin MYB Activators, IbMYB2 and IbMYB3, in Sweetpotato"

_plants, 2025, doi:10.3390/plants14182896_

Round 1

Reviewer 1 Report

Comments and Suggestions for Authors

This manuscript identified and functionally verified two new anthocyanin regulatory factors IbMYB2 and IbMYB3 in sweet potato, which supplemented the existing research of IbMYB1 and made an important contribution to understanding the regulatory network of anthocyanin accumulation in sweet potato. Comments are as follows:

1. The mutation rate (R) used in evolution time (Ka/Ks) directly borrows the rate of CHS gene in Arabidopsis thaliana. Is there any correlation between the mutation rate of sweet potato and its related species? Is the rate of using foreign groups appropriate? Please explain it a little or list it as a limitation of the analysis.

2. Multi-species tissue-specific expression analysis is a great advantage of this study. However, the thermogram only shows the δ CT value. It is suggested to supplement statistical analysis, for example, to mark genes with significant differences in expression between different tissues/varieties with symbols in thermogram. Otherwise, the conclusions of "leading" and "main influence" are slightly subjective.

3. Please indicate the number of biological repeats in each tissue of each variety (n=? ), and the technical duplication of qRT-PCR.

4. The phenotype of IBM Yb 3-ox is very weak (only 5 out of 60 strains appear light purple). Please discuss the possible reasons.

5. Please specify the FDR correction method in DEG screening.

6. The potential molecular mechanism of functional differentiation of IbMYB1/2/3 should be discussed more deeply, such as the difference of promoter elements and the change of protein interaction interface, not just the difference of expression patterns.

7. It is suggested to compare and discuss the MYB gene cluster of sweet potato with similar gene clusters in other reported plants, or to list this as a limitation of analysis.

8. When introducing the biological significance of anthocyanins, we can cite the research in different crops across species, emphasizing the widespread existence and value of anthocyanins in crops, especially the papers published in the past five years, and especially recommend the following two: https://doi.org/10.3389/fpls.2023.1154535; https://doi.org/10.1016/j.indcrop.2025.121100

9. English is fluent as a whole, but there are some long sentences and individual grammatical errors.

10. The repetition rate between manuscripts and published documents is slightly higher (21%), please reduce it to less than 20%.

Comments on the Quality of English Language

English is fluent as a whole, but there are some long sentences and individual grammatical errors.

Author Response

Dear reviewer,
Thank you for your comments. We revised the manuscript as suggested.

Comments 1. The mutation rate (R) used in evolution time (Ka/Ks) directly borrows the rate of CHS gene in Arabidopsis thaliana. Is there any correlation between the mutation rate of sweet potato and its related species? Is the rate of using foreign groups appropriate? Please explain it a little or list it as a limitation of the analysis.

Response 1: The evolutionary rates of genes are influenced by many factors such as species and gene function. Genes with more fundamental functions evolve more slowly, making it challenging to establish a unified standard. Currently, there are no definitive reports on the specific mutation rate (R) MYB genes or the Ipomoea genus. To make a rough estimate, the CHS gene in A. thaliana is commonly used as a standard for estimating evolutionary rates in angiosperms. Additionally, since the CHS gene is also a key player in anthocyanin synthesis pathways, this study adopts the R value of the A. thaliana CHS gene. We added the "approximate” to denote the rough estimation of divergence time (Line 172, 4815) and further explained in the 4.3 section (Line 484-485).

Comments 2. Multi-species tissue-specific expression analysis is a great advantage of this study. However, the thermogram only shows the δ CT value. It is suggested to supplement statistical analysis, for example, to mark genes with significant differences in expression between different tissues/varieties with symbols in thermogram. Otherwise, the conclusions of "leading" and "main influence" are slightly subjective.

Response 2: Thank you for your comments. We have added Figure S6 to illustrate the statistical analysis. In the comparison of gene expression across multiple tissues in various varieties, the 2−ΔΔCt method is not appropriate as no suitable sample can be selected as the CK sample. If a low-expressing sample is chosen, nearly all other samples will exhibit significant differences from it, rendering the comparison invalid. The ΔCt method, relative to the reference genes, was used here. In Figure 4, the heatmap employs a ΔCt interval of 3 (8-fold change), with color gradients roughly indicating significant expression differences ( Line 261, 555).

Comments 3. Please indicate the number of biological repeats in each tissue of each variety (n=? ), and the technical duplication of qRT-PCR.

Response 3: We have the information in section "4.1 Plant materials" (line 446) and added descriptions in section "4.8 qRT-PCR" (line 542). Since each biological replicate was derived from three vines or tubers pooled together, we generally did not perform technical duplication. Technical replicates were only conducted when the results from the three biological replicates varied significantly, though this was not specifically mentioned in the text.

Comments 4. The phenotype of IbMYB 3-ox is very weak (only 5 out of 60 strains appear light purple). Please discuss the possible reasons.

Response 4: We have added discussion about this (Section 3.4 Line 416-419). From transient anthocyanin induction assays in tobacco, IbMYB3 exhibited the strongest anthocyanin induction ability among IbMYB1/2/3, as demonstrated by repetitions of the assay (Figure 3, Line 198). In the transcriptomic data of IbMYB2-ox plants, the inhibitory IbMYB27 was strongly induced. It is speculated that the weak reddish phenotype of IbMYB3-OX may be due to its stronger induction of negative feedback, making it difficult to maintain the red color in fibrous roots. However, the speculations require further verification in subsequent studies.

Comments 5. Please specify the FDR correction method in DEG screening.

Response 5: Revised as suggested (Line 538). The FDR correction method is currently widely used by transcriptome companies, primarily to eliminate false positive results. 

Comments 6. The potential molecular mechanism of functional differentiation of IbMYB1/2/3 should be discussed more deeply, such as the difference of promoter elements and the change of protein interaction interface, not just the difference of expression patterns.

Response 6: Thank you for your suggestion. We added discussion about this (Line 382-392).

Comments 7. It is suggested to compare and discuss the MYB gene cluster of sweet potato with similar gene clusters in other reported plants, or to list this as a limitation of analysis.

Response 7: Thank you for your suggestion. We found an intriguing study indicating that the MYB tandem cluster may have originated in a highly conserved region of numerous eudicot genomes. This was determined through collinear analysis among the genomes of various eudicot species, including A. thaliana, orange, pear, grape, apple, carrot, Chinese bayberry, and Prunus. (Xue et al. 2024; DOI: 10.1093/plphys/kiae063). We added several citations and discussed this (Line 377-381).

Comments 8. When introducing the biological significance of anthocyanins, we can cite the research in different crops across species, emphasizing the widespread existence and value of anthocyanins in crops, especially the papers published in the past five years, and especially recommend the following two: https://doi.org/10.3389/fpls.2023.1154535; https://doi.org/10.1016/j.indcrop.2025.121100

Response 8: Revised as suggested. We have added citations for the two studies to highlight the widespread presence and value of anthocyanins across species (Lines 43-46). And also in the discussion (Line 423)

Comments 9. English is fluent as a whole, but there are some long sentences and individual grammatical errors.

Response 9: Revised as suggested. We conducted a thorough revision of the entire text. We split some long sentences into shorter ones and corrected issues related to grammar, spelling, and punctuation.

Comments 10. The repetition rate between manuscripts and published documents is slightly higher (21%), please reduce it to less than 20%.

Response 10: Revised as suggested. We conducted a paid plagiarism check. The sections with high duplication rates were as follows: (1) Page 1, the author's affiliation and email addresses; (2) Page 2, the list of gene names along the anthocyanin pathway; (3) Page 15, the description of the transcriptional analysis workflow; (4) Page 17, the Fund lists (repeated from papers previous published by other members of our group); (5) Content from the article template, such as the “Copyright” "Data Availability Statement" and "Conflicts of Interest"; (6) long phrases scattered throughout the text (e.g., vector names, website URLs).
The only text that can be modified is the transcriptional analysis workflow provided by the company, and we have removed the routine details to reduce the duplication rate (Line 530-533).

Reviewer 2 Report

Comments and Suggestions for Authors

Sweetpotato varieties display diverse purple pigmentation due to anthocyanin accumulations. While most studies on the underlying MYB activators have focused on IbMYB1 in purple-fleshed tubers, the color diversity suggests involvement of other MYB activators. Here, in this study, the authors explored the chromosomal localization, phylogeny, and evolutionary scenario of IbMYB1/2/3 using four Ipomoea genomes. Their results showed that overexpression of IbMYB1/2/3 in sweetpotato all induced purple fibrous roots. Transcriptomics of IbMYB2-OX fibrous roots revealed the up-regulation of the entire anthocyanin pathway genes, and IbMYB27 and IbHLH2 were the among the most highly up-regulated transcription factors and this inverstigation highlights IbMYB2/3 as fine-tuning sweetpotato’s purple pigmentation, advancing our genetic understanding and offering potential targets for breeding sweetpotato varieties with improved nutritional and aesthetic value. Thus, the research results are original. In general, this manuscript was well-written and results are significant, the conclusions consistent with the evidence and arguments presented in the data sheet and the results addressed the main question. I have the following comments:

Comments:

  1. For the abstract, which should stress the significant and application value of this study;
  2. The title of Figure 3, Functional verification of IbMYB1, IbMYB2, and IbMYB3, the genes should be itatic;
  3. Figure 5, overexpression of IbMYB1, IbMYB2, and IbMYB3 induces anthocyanin accumulation in sweetpotato roots. I think the authors should measure the anthocyanin accumulation or check the key genes expression in synthesis of anthocyanin accumulation?
  4. There is no references were cited from 2025, thus the references need to be updated.
  5. In the conclusion part, the authors should stress the significant and application value of this study.

Author Response

Dear reviewer,
Thank you for your comments. We revised the manuscript as suggested. 

Comments 1: For the abstract, which should stress the significant and application value of this study;
Response 1: Revised as suggested. We revised the sentences at the end of the abstract (Line 31-33). 

Comments 2: The title of Figure 3, Functional verification of IbMYB1, IbMYB2, and IbMYB3, the genes should be itatic;
Response 2: Revised as suggested. 

Comments 3: Figure 5, overexpression of IbMYB1, IbMYB2, and IbMYB3 induces anthocyanin accumulation in sweetpotato roots. I think the authors should measure the anthocyanin accumulation or check the key genes expression in synthesis of anthocyanin accumulation?
Response 3: Thank you for your concern. As the purple phenotype complements anthocyanin coloration well, we prioritized the precious purple fibrous root materials for transcriptomic and qPCR analyses. For IbMYB2-OX roots, we conducted transcriptomic analysis and qRT-PCR screening of all anthocyanin pathway genes (Figures 6E,7C). For IbMYB1-OX and IbMYB3-OX roots, we conducted qRT-PCR for several anthocyanin pathway genes (data not shown). All the results were consistent with up-regulation of anthocyanins. 
However, we neglected the routine experiment of anthocyanin determination, which is a limitation of our study, and we will pay attention to this in future studies.

Comments 4: There is no references were cited from 2025, thus the references need to be updated.
Response 4: Revised as suggested. We added several references from 2023 to 2025, including two from 2025. Now, a total of 20 references are from 2020 to 2024 among the 37 references.

Comments 5: In the conclusion part, the authors should stress the significant and application value of this study.
Response 5: Revised as suggested. We revised the sentences at the beginning and end of the conclusion section. 

Reviewer 3 Report

Comments and Suggestions for Authors

Sweetpotato varieties exhibit diverse purple pigmentation due to anthocyanin accumulation. The authors investigated the chromosomal localization, phylogeny, and evolutionary scenario of the anthocyanin MYB activator IbMYB1/2/3. Using methods such as tobacco transient induction, promoter activation experiments, multi-variety expression analysis, sweetpotato overexpression, and transcriptomic analysis, they elucidated its functional role in anthocyanin synthesis. This manuscript reveals the precise regulatory role of IbMYB1/2/3 in purple pigment deposition in sweet potatoes, providing deeper insights into the complex regulation of anthocyanin accumulation in sweet potato tissues. This manuscript has clear research objectives, comprehensive research content, and an accurate analysis of results. There are some comments:

  1. The introduction needs to be expanded, such as the regulatory role of IbMYB1/2/3 in anthocyanin synthesis in sweetpotatoes or other plant varieties.
  2. The significance of IbMYB1/2/3 in anthocyanin synthesis in sweetpotatoes should be added to the introduction.
  3. Appropriate references should be added to the Materials and Methods section.
  4. Details are needed in the “Materials and Methods” section, such as specifying the absorption wavelength range and emission wavelength range when describing subcellular localization.
  5. The reagent kit information used in the manuscript needs to be labeled with the number, city, and manufacturer.
  6. Line 526. Why choose two housekeeping genes?
  7. Line 253, “presented as ΔCt values relative to housekeeping genes”. I am unclear on how the housekeeping gene is used to calculate gene expression levels, and whether Actin or ARF is used as a reference. Please add this information to the manuscript.
  8. In Figure 5, in addition to being described in the figure caption, the scale should also be added to the corresponding image.
  9. Add paragraphs with clear data statistics to the manuscript.
  10. The dates used for analysis in each database should be noted in the manuscript.
  11. The discussion section needs to be further strengthened based on the data in the manuscript. The current discussion is insufficient.
  12. Please check the spelling and punctuation throughout the manuscript. Such as line 340, line 431, and line 452.

Author Response

Dear reviewer,
Thank you for your comments. We revised the manuscript as suggested. 

Comments 1: The introduction needs to be expanded, such as the regulatory role of IbMYB1/2/3 in anthocyanin synthesis in sweetpotatoes or other plant varieties.
Response 1: Thank you for your suggestion. We have restructured the introduction and provided an explanation on the MYB1/2/3 status in well-studied ornamental morning glories (lines 87-89).

Comments 2: The significance of IbMYB1/2/3 in anthocyanin synthesis in sweetpotatoes should be added to the introduction.
Response 2: Thank you for your suggestion. In the introduction, we detailed the history of the discovery of numerous variants of IbMYB1 and IbMYB2/3 across two paragraphs. We have now reorganized the introduction to better address your question.

Comments 3: Appropriate references should be added to the Materials and Methods section.
Response 3: Revised as suggested. 

Comments 4: Details are needed in the “Materials and Methods” section, such as specifying the absorption wavelength range and emission wavelength range when describing subcellular localization.
Response 4: Revised as suggested. Due to the length of this article, routine experiments have been concisely described, while key experiments, such as sweetpotato transformation, have been detailed extensively. The subcellular localization section has been enhanced with the addition of four sets of fluorescence channel excitation light and emission data (Lines 527-530).

Comments 5: The reagent kit information used in the manuscript needs to be labeled with the number, city, and manufacturer. 
Response 5: Revised as suggested. Due to the high repetition rate in the transcriptome sequencing workflow, details were removed. Other kits have been labeled.

Comments 6: Line 526. Why choose two housekeeping genes?
Response 6: Conventional housekeeping genes frequently exhibit insufficient stability across species, and many journals (e.g. Frontiers in plant science, and many higher journals) now mandate the use of two housekeeping genes for qRT-PCR. Here, we chose actin and the commonly used ARF gene for sweetpotato. In our assay, ARF was demonstrated to be more reliable than actin.

Comments 7: Line 253, “presented as ΔCt values relative to housekeeping genes”. I am unclear on how the housekeeping gene is used to calculate gene expression levels, and whether Actin or ARF is used as a reference. Please add this information to the manuscript.
Response 7: The average Ct values of Actin and ARF were used for correction. Additional information is provided in the Methods section (lines 547).

Comments 8: In Figure 5, in addition to being described in the figure caption, the scale should also be added to the corresponding image.
Response 8: Revised as suggested. 

Comments 9: Add paragraphs with clear data statistics to the manuscript.
Response 9: Data statistics varied among assays and we previously present the information at the at the end of the figure legends. Now, we revised the method description, especially in “4.8. qRT-PCR”, to show more clear presentation (Line 538,550-552, 555-556, 558 ).

Comments 10: The dates used for analysis in each database should be noted in the manuscript.
Response 10: Revised as suggested. Section 4.2 (Line 460-470)

Comments 11: The discussion section needs to be further strengthened based on the data in the manuscript. The current discussion is insufficient. 
Response 11: Revised as suggested. We have strengthened the discussion in several places, including the tandem gene clusters (Line377-381), the functional differentiation (Line382-392), and the negative feedback caused by overexpression of MYB1/2/3 (Line417-419).

Comments 12: Please check the spelling and punctuation throughout the manuscript. Such as line 340, line 431, and line 452. 
Response 12: Thank you for your careful check. We conducted a thorough revision of the entire text. We split some long sentences into shorter ones and corrected issues related to grammar, spelling, and punctuation.

Round 2

Reviewer 1 Report

Comments and Suggestions for Authors

Congratulations. The comments have been well resolved, and it is recommended to accept in present form.

Reviewer 3 Report

Comments and Suggestions for Authors

The author has responded to all comments and made revisions to the manuscript. The current manuscript can be accepted for publication.